# Applicability of the inverse dispersion method to measure emissions from animal housings

**Marcel Bühler**[1,2]**, Christoph Häni**[1]**, Albrecht Neftel**[3]**, Patrice Bühler**[1]**, Christof Ammann**[4]**, and Thomas Kupper**[1]

[1]School of Agricultural, Forest and Food Sciences HAFL, Bern University of Applied Sciences, 3052 Zollikofen, Switzerland
[2]Department of Biological and Chemical Engineering, Aarhus University, 8000 Aarhus, Denmark
[3]Neftel Research Expertise, 3033 Wohlen bei Bern, Switzerland
[4]Climate and Agriculture Group, Agroscope, 8046 Zürich, Switzerland

**Correspondence:** Marcel Bühler (mb@bce.au.dk)

**Abstract.** Emissions from agricultural sources substantially contribute to global warming. The inverse dispersion method (IDM) has been successfully used for emission measurements from various agricultural sources. The IDM has also been validated in multiple studies with artificial gas releases mostly in open fields. Release experiments from buildings have rarely been conducted and were partly affected by additional nearby sources of the target gas. Specific release studies for naturally ventilated animal housings are lacking. In this study, a known and predefined amount of methane ($CH_4$) was released from an artificial source inside a barn that mimicked a naturally ventilated dairy housing, and IDM recovery rates, using a backward Lagrangian stochastic (bLS) model, were determined. For concentration measurements, open-path devices (OPs) with a path length of 110 m were placed in a downwind direction of the barn at fetches of $2.0h$, $5.3h$, $8.6h$, and $12h$ ($h$ equals the height of the highest obstacle), and a 3D ultrasonic anemometer (UA) was placed in the middle of the first three OP paths. Upwind of the barn, an additional OP and a UA were installed. The median IDM recovery rates determined with the UA placed upwind of the barn and the downwind OP ranged between 0.55–0.75. It is concluded that, for the present study case, the effect of the building and a tree in the main wind axis led to a systematic underestimation of the IDM-derived emission rate probably due to deviations in the wind field and turbulent dispersion from the underlying assumptions of the used dispersion model.

## 1 Introduction

The growth in atmospheric methane ($CH_4$) concentration is largely due to emissions from the fossil fuel, agriculture, and waste sectors (Arias et al., 2021). For the period 2008–2017, global $CH_4$ emissions from agriculture and waste management contributed 56 % of the total anthropogenic $CH_4$ emissions (Saunois et al., 2020). Within the livestock sector at a global scale, $CH_4$ mainly originates from enteric fermentation in the digestive tract of ruminants and to a minor extent from emissions from manure management (Gerber et al., 2013). A common housing system for cattle is loose housing in naturally ventilated buildings (Sommer et al., 2013). To improve national emission inventories and test mitigation effects under real-world conditions, accurate measurements are necessary. For confined sources of greater complexity, the inverse dispersion method (IDM) has become established in recent years. The IDM is a micrometeorological method that combines the measurement of the concentration enhancement downwind of the spatially defined source with an atmospheric dispersion model. For agricultural emissions, most often the backward Lagrangian stochastic (bLS) model approach by Flesch et al. (1995) is used. This bLS model has been verified in multiple release experiments on open fields that reflected ideal conditions in terms of the Monin–Obukhov similarity theory (Flesch et al., 2004). Ideal conditions for the bLS model are a horizontally homogeneous surface layer and a distance between source and sensors of less than 1 km (Flesch et al., 1995). Also, under less ideal conditions in terms of the Monin–Obukhov similarity theory, the

bLS model showed its aptitude for a wide range of sources (e.g. Bühler et al., 2022, 2021; Flesch et al., 2009; Laubach et al., 2013; VanderZaag et al., 2014). However, there are only a few studies available where the gas was released within or close to a building or structure. Baldé et al. (2016) and Hrad et al. (2021) released $CH_4$ at real-world facilities in addition to the $CH_4$ from sources existing at the sites. McGinn et al. (2006) conducted a release experiment at a barn with three release positions on top of the roof and three positions outside the walls of the barn. Gao et al. (2010) released $CH_4$ via four side vents of a barn. The barn in the study of Gao et al. (2010) is comparable to a mechanically ventilated building which is common for fattening pigs or poultry.

In this study, we present an experiment with the artificial release of $CH_4$ within a building similar to a naturally ventilated dairy housing. The goal of this experiment was to test the IDM with bLS modelling for the quantification of emissions from an agricultural building with natural ventilation under conditions that were as realistic as possible. Compared to Gao et al. (2010), multiple 3D ultrasonic anemometers (UAs) were available in our experiment. Thus, the focus was on the positioning of the open-path concentration sensors and the ultrasonic anemometers at different horizontal distances downwind of the source.

## 2 Material and methodology

### 2.1 Experimental site and periods

The release experiment was conducted in a barn located in the Central Plateau of Switzerland (47.04307° N, 7.22691° E). The barn allowed a setup which mimicked a naturally ventilated dairy housing. About 350 m northeast of the barn was a river with 4 m high dams on each side and trees that were about 25 m high. There were no other obstacles between the barn and the dam. The canopy height directly around the barn was 20 cm and lower, and it remained constant over the course of the measurements. The barn was 25 m long, 17 m wide, and 7 m high (Fig. 1). During the release experiment, about 17 % of the barn's surface was occupied by storage boxes stacking almost up to the ceiling. Despite other agricultural equipment inside the barn, about 33 % of the south end of the barn was empty. The barn had a 4.8 m wide and 4.0 m high gate on each transverse side. During the $CH_4$ releases, the gate on the south side was fully open, whereas the gate on the north side was opened 1.3 m. The north-facing wall of the barn was impermeable; however, the south wall and the longitudinal side walls exhibited small holes and cracks all over, allowing air exchange through the wall. At both longitudinal sides of the barn there were gaps of about 0.6 m below the roof which were covered by cracked plastic sheets. About 20 m southwest of the barn was a tree of about 15 m height (Fig. 1).

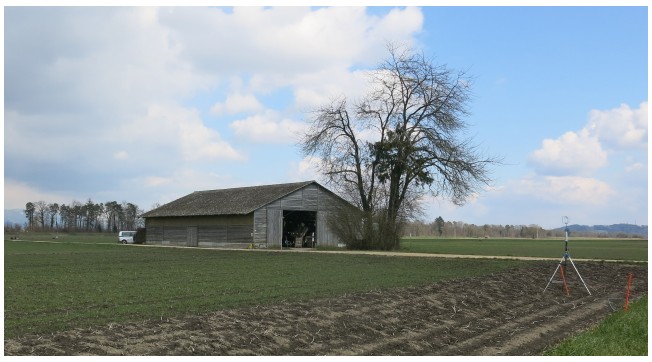

**Figure 1.** Barn used for the $CH_4$ release experiment and the adjacent tree. In the background, a river dam with trees on it is visible. The photo was taken from the southwest side of the barn. In the foreground is a 3D ultrasonic anemometer (UA-2.0*h*).

A petrol-powered generator (Honda EU20i), located outside the barn at the southeast side, provided the necessary power for all the instruments.

The wind and concentration measurements lasted over several weeks from 5 to 26 March 2021. A first intercomparison (IC1) of the open-path devices was conducted from 5 to 10 March 2021. The measurement campaign (MC) took place from 18 March 2021 11:00 LT to 21 March 2021 13:00 LT (UTC+1). Within this MC, $CH_4$ was released with a $CH_4$ source inside the barn from 19 March 2021 10:30 LT to 19 March 2021 16:40 LT, denoted as daytime release, and 19 March 2021 21:50 LT to 20 March 2021 06:50 LT, denoted as nighttime release. The second intercomparison (IC2) of the open-path devices was conducted from 21 March 2021 15:00 LT to 26 March 2021 10:00 LT. $CH_4$ was also released during part of IC2 (Sect. S1 in the Supplement).

### 2.2 Methane source

For the release during the MC and half of IC2, a gas bundle of 12 cylinders with 50 L at 200 bar, each with high-purity (> 99.5 % mol) $CH_4$, was used to supply the $CH_4$ source. For the rest of the release in IC2, one gas cylinder with 50 L at 200 bar was used. Attached to the bundle was a pressure regulator (Fig. 2). The pressure on the high-pressure side was measured with a digital pressure sensor (LEX1-Ei 200 bar 81770.5, KELLER AG, Winterthur, Switzerland). The low-pressure side was set to 3 bar. The pressure regulator and the mass flow controller (MFC; EL-FLOW Select F-202AV-M20-AGD-22-V, Bronkhorst High-Tech BV, Ruurlo, the Netherlands) were connected by polyethylene naphthalate (PEN) tubing (Festo, PEN-16X2,5-BL-100 551449) with an inner diameter of 10.8 mm. After the MFC, there was an 8 m long PEN tube with an inner diameter of 10.8 mm connected to a gas distribution block made of aluminium with three outlets (ITV, 124 A24 G1/2″). Each outlet had an L-fitting (Festo, QSL-G1/2-16 186126) and 1.5 m of the

same tubing connected to another gas distribution block with eight outlets, with a reduction in the tubing diameter to 2.7 mm (Festo, FR-8-1/4 2078). To each of these outlets an L-fitting (Festo, QSLL-1/4-4 190662) and a 20 m long tube with an inner diameter of 2.7 mm (Festo, PPEN-4X0,75-BL-500 551444) were attached to release $CH_4$. At the end of these tubes, no pressure reduction was added. The total pressure drop of the system was expected to be around 0.4 bar.

The pressure and the temperature recorded with the KELLER pressure sensor were logged with 10 Hz (Sect. S2). From the MFC, the set-point ($L_n$ min$^{-1}$), the flow rate ($L_n$ min$^{-1}$), and the temperature were logged with 0.1 Hz resolution. The MFC had a maximum flow of 160 $L_n$ min$^{-1}$ and was calibrated for $CH_4$ at 15 °C. During IC2, the set-point was varied between 50 and 160 $L_n$ min$^{-1}$, whereas the flow was kept constant at 140 $L_n$ min$^{-1}$ during the MC (Sect. S2). 140 $L_n$ min$^{-1}$ corresponds to 6.02 kg $CH_4$ h$^{-1}$, which represents an emission rate of about 360 dairy cows. This emission rate was chosen to achieve sufficient concentration enhancement at the concentration measurement locations and is thus an adequate signal-to-noise ratio. Under Swiss regulations, the space in the barn would be insufficient for 360 dairy cows. The cumulative flow through the MFC whilst the gas bundle was connected was within 1 % of the $CH_4$ volume inside the gas bundle.

The gas bundle and the MFC were placed outside the barn on the north side. The release points inside the barn were at 1.5 m above ground. The 24 release points were equally distributed in the southern half of the barn.

At the beginning of the daytime release, a short circuit caused a shutdown of the power generator for about 30 min. On 20 March 2021 around 01:00 LT (nighttime release), the computer was needed to check data from a UA; thus, the $CH_4$ release was stopped for a few minutes.

### 2.3 Methane concentration measurements

The $CH_4$ concentrations were measured with five GasFinder3-OP (Boreal Laser Inc., Edmonton, Canada) open-path tunable diode laser absorption spectrometers (hereafter denoted as OPs). Mirrors with 12-corner cubes were used as retroreflectors. Data with insufficient light intensity were removed. Device-specific relationships determined by factory calibration were applied to the measured concentration using local air temperature and air pressure measured by a weather station (WS700-UMB Smart Weather Sensor, G. Lufft Mess- und Regeltechnik GmbH, Fellbach, Germany) placed about 100 m southwest of the barn (Fig. 3). The measured $CH_4$ concentrations (0.3–1 Hz resolution) were averaged to 10 min periods, and periods with data coverage lower than 75 % (7.5 min) were removed. The concentrations between the five OPs were intercalibrated with data from the parallel measurements in IC1 and IC2 and were corrected for slope and offset using linear regression. Afterwards, an additional offset correction was applied

based on periods during the MC when no $CH_4$ was released. The precision for the employed OP was determined from the parallel measurements according to Häni et al. (2021; Sect. S3).

### 2.4 Turbulence measurements and data filtering

Four 3D UAs (Gill WindMaster, Gill Instruments Ltd., Lymington, UK) were used to determine turbulence parameters. A two-axis coordinate rotation was applied to the wind vector rotation. From the 10 Hz data, 10 min periods were built.

As the bLS model uses Monin–Obukhov similarity theory scaling, the UA data required compatibility with Monin–Obukhov similarity theory assumptions and, consequently, a screening of data with the goal to exclude situations that substantially deviated from these assumptions. The goal of this screening or quality-filtering was to retain as much data as possible without introducing too many erroneous results. Quality filters were applied for the wind direction and the friction velocity $u_*$. Data with $u_* \leq 0.15$ m s$^{-1}$ were excluded (Flesch et al., 2005b). The wind direction intervals are given in Sect. S4. No additional quality filters were applied.

### 2.5 Experimental setup

For all measurements, the five OPs (sensor modules and retroreflectors) were placed 1.60 m above ground level with a path length of 110 m. In IC1, the OPs were placed about 100 m southwest of the barn. During the MC, four OPs were placed southwest of the barn, and one was placed northeast of the barn (Fig. 3). The distances between the barn and the middle of the OP paths on the southwest side were 50, 100, 150, and 200 m. Since the tree located 20 m southwest of the barn was the highest obstacle in the experiment, the locations of the instruments (OP and UA) are indicated as the relative distance to the tree (multiple of the tree height $h = 15$ m), resulting in fetches of $2.0h$, $5.3h$, $8.6h$, and $12h$. Three UAs were placed downwind in the middle of the OP paths, and one was placed upwind of the barn. The distance between the upwind UA (UA-UW) placed in the northeast of the barn and the trees on the dam in the direction of 52° (mean wind direction during release) was 370 m, which corresponded to a distance of $> 12h$ when regarding the trees on the dam as dominant height. The measuring height of all UAs was 2.16 m above ground level. For IC2, all five OPs were placed next to each other about 50 m southwest of the barn, and one UA was placed 55 m southwest of the barn at 2 m above ground level (Sect. S5).

### 2.6 Inverse dispersion method

A backward Lagrangian stochastic model (Flesch et al., 1995, 2004) was used to establish the relationship between the emissions at the source and the concentration mea-

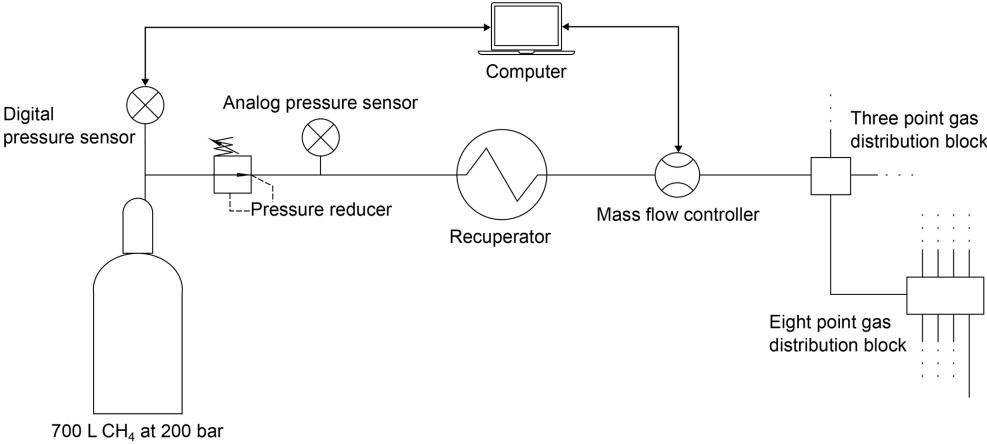

**Figure 2.** Schematic of the $CH_4$ source for the artificial release experiment.

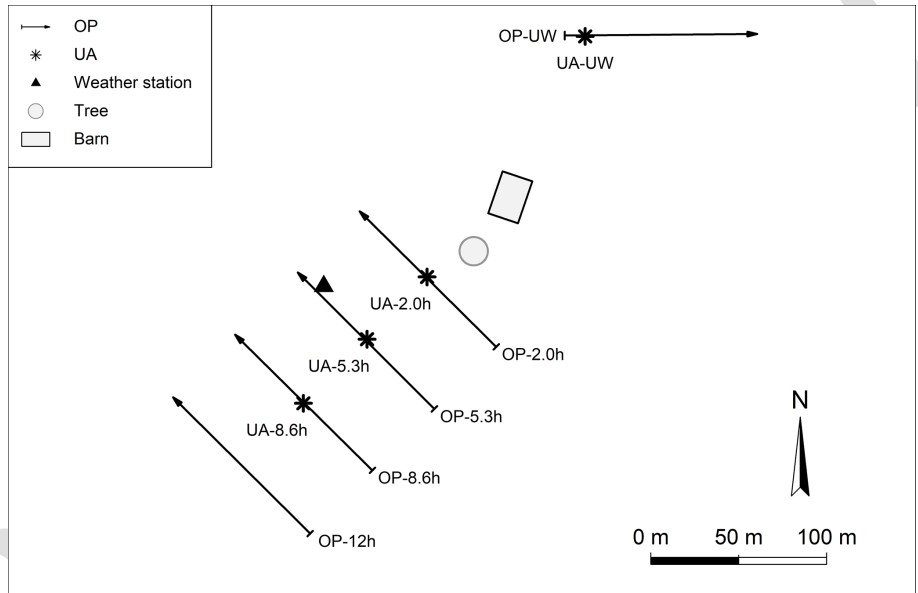

**Figure 3.** Schematic overview of the measurement setup during the measurement campaign. OP: open-path device. UA: 3D ultrasonic anemometer. UW: upwind. The numbers behind the OP and UA represent the fetch.

sured downwind of the source under investigation. This concentration–emission relationship is quantified by the dispersion factor $D$ $(\mathrm{s\,m^{-1}})$, which depends both on the geometrical configuration of the source and concentration sensor and on the turbulence and the wind field. To separate the contribution of the source from the incoming (background) concentration at the downwind measurement location, the concentration upwind of the source is also measured. With the area $A$ $(\mathrm{m^{-2}})$ of the source, the emission of the source $Q$ $(\mathrm{kg\,s^{-1}})$ can be calculated (Eq. 1):

$$Q = \frac{C_{\mathrm{DW}} - C_{\mathrm{UW}}}{D} \cdot A, \tag{1}$$

where $C_{\mathrm{UW}}$ and $C_{\mathrm{DW}}$ are the upwind (background) and downwind concentration $(\mathrm{kg\,m^{-3}})$.

The bLS model by Flesch et al. (2004) uses Monin–Obukhov similarity theory formulas to specify turbulence statistics in the inertial sublayer of the atmosphere that are derived from the friction velocity, the Obukhov length, and the roughness length measured by the UA. Monin–Obukhov similarity theory needs stationarity and homogeneity regarding the turbulence conditions; therefore, the measurement site should be horizontally homogeneous and flat over a large area. Additionally, the bLS model assumes a homogeneous diffusive ground source. A building or a structure violates these conditions; thus, based on experimental field trials, it is recommended that the distance between the source and the downwind measurement locations should be not less than 10 times the source height so that the turbulence fulfils the as-

sumptions of homogeneity and stationarity (Gao et al., 2010; Harper et al., 2011).

The OPs in the bLS model were approximated by a series of point sensors with 1 m spacing along the path length. For each of these point sensors and each emission interval, 1 million backward trajectories were used to calculate the value of $D$. The simulations were run in R statistical software (v3.6.6; R Core Team, 2019) using the package bLSmodelR (Häni et al., 2018) available at https://github.com/ChHaeni/bLSmodelR (Häni, 2021). The following quantities were used as input parameters for the bLS model: the coordinates of the source (barn area) and the OP, the height of the OP and the UA above ground, the friction velocity, the Monin–Obukhov length, the roughness height, the wind direction, the standard deviation of the wind direction, the displacement height, and the standard deviation of the $u$, $v$, and $w$ wind divided by the friction velocity.

## 3 Results

A general overview of the weather conditions during the measurement campaign is given in Fig. 4. Due to a change in wind direction, the $CH_4$ release was stopped for several hours until the conditions were suitable again. During the daytime release, the inverse of the Monin–Obukhov length ($L^{-1}$) recorded by the UA-UW was between 0 and $-0.1\,\mathrm{m^{-1}}$; thus the atmospheric conditions were moderately unstable. During the nighttime release, the atmospheric conditions were moderately stable, with $L^{-1}$ between 0 and $+0.1\,\mathrm{m^{-1}}$ (Fig. 5, Table 3). The mean wind direction, the mean wind speed, and the mean friction velocity recorded by the UA-UW in the MC during the $CH_4$ release phases were $51.7°$, $3.5\,\mathrm{m\,s^{-1}}$, and $0.28\,\mathrm{m\,s^{-1}}$, respectively (Table 3).

### 3.1 Concentration measurements

The precision of OP concentration measurements, expressed as path-integrated concentration in parts per million metre (ppm m), reflects the concentration integrated over the single distance between the sensor module and the retroreflector. It was determined during the IC1, MC, and IC2 and ranged between 3.3 `TS1` and 8.5 ppm m (Table 1). `CE1` During the MC, only periods 10 min before and 60 min after a $CH_4$ release were used to determine the precision. The precision of the OP was lowest during the MC. The median concentration enhancements ($\Delta C = C_{DW} - C_{UW}$) for the daytime and nighttime release are given in Table 2. For these $\Delta C$ values, we only used periods for which recovery rates were also determined. The concentration enhancements were higher during the nighttime release when the atmospheric conditions were stable than during the daytime release with unstable conditions.

**Table 1.** Precision of the OP determined according to Häni et al. (2021). $N$ is the number of intervals used to determine precision.

|  |  | IC1 | MC | IC2 | All data |
|---|---|---|---|---|---|
| OP-2.0$h$ | Precision (ppm m) | 4.3 | 5.8 | 5.9 | 5.3 |
|  | $N$ | 695 | 232 | 504 | 1431 |
| OP-5.3$h$ | Precision (ppm m) | 3.3 `TS2` | 8.5 | 4.4 | 4.4 |
|  | $N$ | 638 | 226 | 505 | 1369 |
| OP-8.6$h$ | Precision (ppm m) | 5.1 | 7.1 | 5.2 | 5.7 |
|  | $N$ | 685 | 228 | 512 | 1425 |
| OP-12$h$ | Precision (ppm m) | 4.3 | 6.3 | 4.3 | 4.8 |
|  | $N$ | 679 | 226 | 286 | 1191 |

### 3.2 Recovery rates

IDM emissions (Eq. 1) and corresponding recovery rates (IDM emissions divided by actual emissions according to gas release) were determined with the different downwind OP instruments using turbulent parameters determined with the UA-UW. During the $CH_4$ release, the data loss due to quality filtering was 8 %, 11 %, 29 %, and 36 % for OP-2.0$h$, OP-5.3$h$, OP-8.6$h$, and OP-12$h$, respectively. The resulting recovery rates were always below 1 (Fig. 5). The median recovery rates for the daytime release during unstable atmospheric conditions ranged between 0.57 and 0.61. For the nighttime release during stable atmospheric conditions, the range was 0.55–0.75 (Table 2). The recovery rates for the nighttime release slightly increased with the distance from the OP to the barn and the adjacent tree, whereas for the daytime release no clear pattern is visible. The highest recovery rates were achieved under stable atmospheric conditions with the OP furthest away from the source.

### 3.3 Influence of the barn and the tree on the wind field

The wind directions of the downwind UA instruments showed systematic deviations from the UA-UW for the wind sector between 40 and 65°, with a maximum deviation at around 55° (Fig. 6). The barn was located 45° of the downwind UA locations (Fig. 3). The closer the downwind UA was placed to the barn, the further the local wind direction deviated towards north from the wind direction measured by the UA-UW (Table 3). A similar pattern was found for the friction velocity (Sect. S6). The observed atmospheric stability was very similar for all UAs (Table 3). Emission recovery rates determined with the UA placed downwind of the barn can be found in the Supplement (Sect. S7).

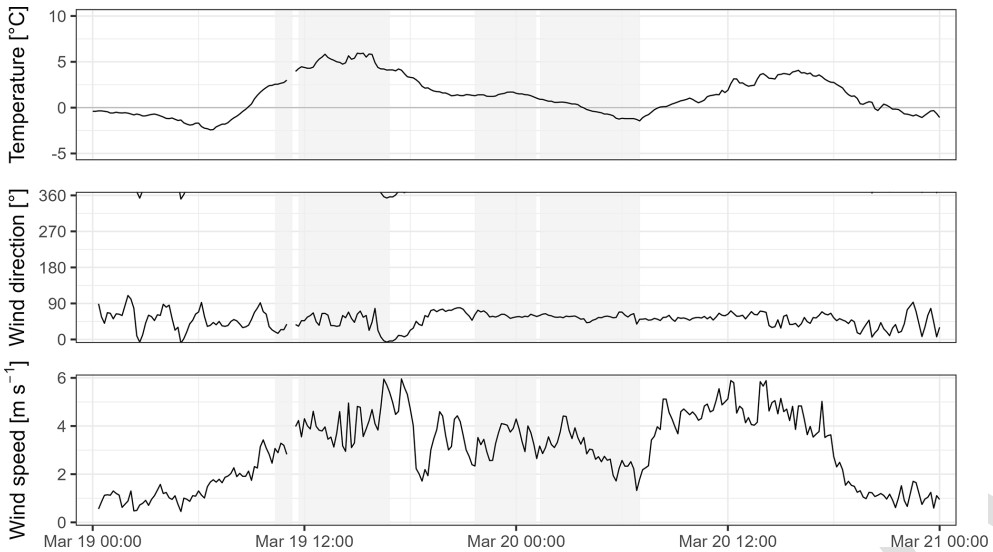

**Figure 4.** Weather conditions as 10 min averages measured with the on-site weather station (temperature) and the UA-UW (wind direction and wind speed) during the measurement campaign. The grey shaded areas indicate the times during which $CH_4$ was released.

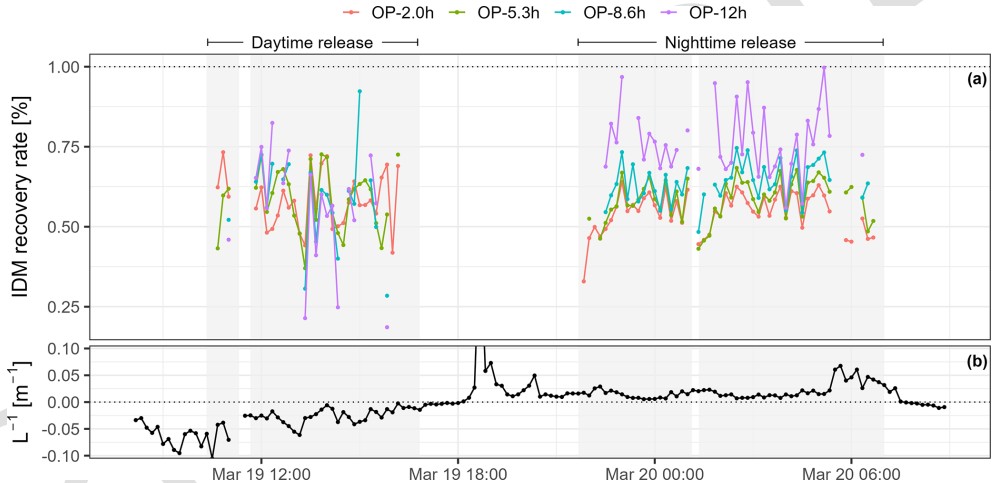

**Figure 5. (a)** Recovery rate for the measurement campaign. The colours indicate the OP used to calculate the recovery rate. **(b)** Atmospheric stability recorded with the UA-UW. Grey shaded areas are the times during which $CH_4$ was released. The time zone is LT (UTC+1).

## 4 Discussion

### 4.1 Influence of the barn on the wind field

The influence of the barn and the tree on the measured downwind turbulence is clearly visible (Fig. 6, Tables 2 and 3). The closer the UA was placed to the tree and the barn, the larger the influence was. The wind must flow around the barn and the tree; thus the largest deviation in the wind direction was measured at around 55°, which corresponds to the southwest–northeast diagonal of the barn. The wind field deviation is also visible in all other turbulence parameters (Sect. S6). The wind speed of the downwind UA closest to the barn was, depending on the wind direction, either higher or lower than the wind speed measured by the UA-UW (Sect. S8). At UA-5.3$h$ and UA-8.6$h$, the wind speed and friction velocity were on average slightly higher than at the upwind location, independent of the atmospheric stability. The difference in the turbulence parameters indicates that all the downwind UAs (fetch between 2.0$h$–8.6$h$) were still in the wake of the barn and the tree. This wind field deviation most likely led to a deviation in the actual emission plume dispersion from the simulations by the bLS model, but, unfortunately, the resulting deviation in the calculated IDM emissions cannot be quantified and corrected for.

**Table 2.** Median recovery rates with standard deviation, median concentration enhancements ($\Delta C$) with standard deviation, and number of 10 min intervals ($N$) for all OPs using the data from the UA-UW for the two releases in the MC. Daytime release (unstable atmospheric conditions) and nighttime release (stable atmospheric conditions).

| OP | | Daytime release ($L < 0$) | Nighttime release ($L > 0$) | Entire MC |
|---|---|---|---|---|
| OP-2.0$h$ | Recovery rate | $0.57 \pm 0.09$ | $0.55 \pm 0.06$ | $0.56 \pm 0.07$ |
| | $\Delta C$ (ppm m) | $58.4 \pm 12.9$ | $78.3 \pm 16.9$ | $70.7 \pm 17.8$ |
| | $N$ | 30 | 50 | 80 |
| OP-5.3$h$ | Recovery rate | $0.61 \pm 0.10$ | $0.59 \pm 0.06$ | $0.59 \pm 0.08$ |
| | $\Delta C$ (ppm m) | $33.2 \pm 8.4$ | $49.8 \pm 12.3$ | $43.5$ TS3 $\pm 14.9$ |
| | $N$ | 29 | 48 | 77 |
| OP-8.6$h$ | Recovery rate | $0.61 \pm 0.15$ | $0.64 \pm 0.06$ | $0.63 \pm 0.10$ |
| | $\Delta C$ (ppm m) | $21.2 \pm 4.8$ | $36.1 \pm 10$ | $30.4 \pm 11.7$ |
| | $N$ | 20 | 42 | 62 |
| OP-12$h$ | Recovery rate | $0.57 \pm 0.18$ | $0.75 \pm 0.10$ | $0.71 \pm 0.17$ |
| | $\Delta C$ (ppm m) | $12.9 \pm 4.4$ | $27.6 \pm 8.6$ | $22.1 \pm 10.9$ |
| | $N$ | 19 | 37 | 56 |

**Table 3.** Mean wind direction (WD), mean wind speed (WS), mean friction velocity ($u_*$), and the mean of the inverse of the Obukhov length ($L^{-1}$) recorded by the UA during the two release phases in the MC.

| | Daytime release | | | | Nighttime release | | | |
|---|---|---|---|---|---|---|---|---|
| | WD ($°$) | WS (m s$^{-1}$) | $u_*$ (m s$^{-1}$) | $L^{-1}$ (m$^{-1}$) | WD ($°$) | WS (m s$^{-1}$) | $u_*$ (m s$^{-1}$) | $L^{-1}$ (m$^{-1}$) |
| UA-UW | 43.0 | 4.0 | 0.32 | $-0.03$ | 58.1 | 3.2 | 0.25 | 0.02 |
| UA-2.0$h$ | 41.1 | 3.9 | 0.31 | $-0.03$ | 50.4 | 2.7 | 0.22 | 0.05 |
| UA-5.3$h$ | 43.9 | 4.2 | 0.36 | $-0.02$ | 55.9 | 3.2 | 0.30 | 0.02 |
| UA-8.6$h$ | 44.5 | 4.3 | 0.37 | $-0.02$ | 55.4 | 3.5 | 0.29 | 0.02 |

## 4.2 Quality filtering and data loss

In this study, a minimum of quality filters was applied. Compared to other measurement campaigns conducted in Switzerland (Bühler et al., 2022, 2021), this campaign was of shorter duration and the atmospheric conditions were more favourable for emission quantification. Additional filters, other than $u_*$ and the wind direction, were tested but not applied in the final analysis, since they excluded more data but did not alter the findings and the mean and median recovery rates.

## 4.3 Recovery rates

The recovery rates of the IDM emission results slightly increased with the distances of the downwind OP to the barn for the nighttime release. For both release phases and for all fetches (2.0$h$–12$h$), the median recovery rate did not exceed 0.75. Gao et al. (2010), who released CH$_4$ via side vents in a barn, achieved a recovery rate of 0.66 for a fetch of 5$h$. However, for a fetch of 10$h$ to 25$h$, Gao et al. (2010) measured recovery rates between 0.93 and 1.03. McGinn et al. (2006) determined recovery rates between 0.59 and 1.05 with an average of 0.86 for a fetch of 9$h$ and larger. For a fetch of 10$h$ at a biogas plant, Hrad et al. (2021) measured a recovery rate of 1.19 (variant 1, USA A). Baldé et al. (2016) released CH$_4$ on the slurry surface of two storage tanks and got recovery rates ranging from 0.91 to 1.20 with an average of 1.05. However, it was unfortunately not possible to determine a fetch from the given data. Among the abovementioned studies, the one by Gao et al. (2010) is the most comparable to our study.

The minimum fetch of 10$h$ often proposed and used for the positioning of downwind concentration measurements might not always be sufficient. In our study, at a fetch of 12$h$, the observed recovery rates showed a median of 0.57 and 0.75 and were never above 1. This suggests that the fetch of 12$h$ from the flow-disturbing obstacle was not enough in this case. However, from the available data in our study, it is not possible to state that longer fetches would considerably improve the recovery rate. This is in line with the statement of Flesch et al. (2005a) that the determination of a universal distance for a fetch that reliably avoids wind disturbances is unlikely.

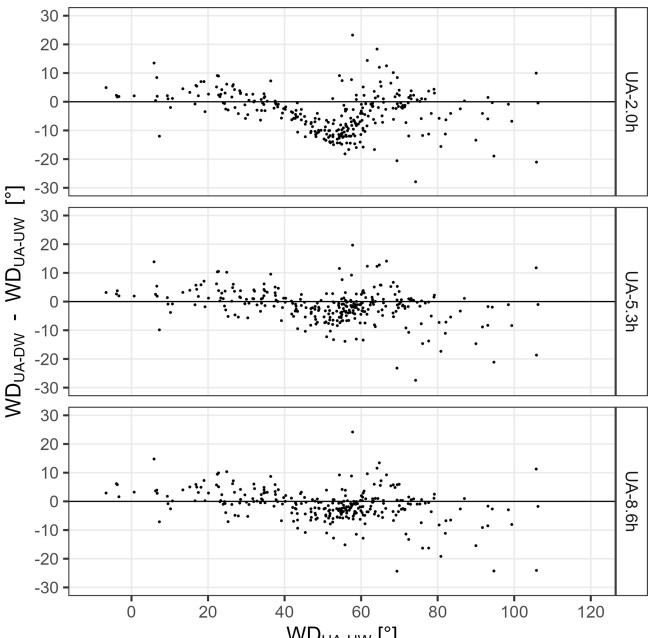

**Figure 6.** Absolute difference in the wind direction between the three downwind UAs (UA-DW) and the upwind UA (UA-UW) recorded during the entire measurement campaign, given as 10 min data. The exact locations of the UAs are given in Fig. 3.

Nevertheless, we tried to rule out or explain possible mechanisms for the low recovery rates. A bias in the release rate that could explain them is unlikely, as the cumulative flow through the MFC was within 1 % of the $CH_4$ volume inside the gas bundle. The precision of the OPs was comparable to the values presented by Häni et al. (2021) for intercomparisons with similar path lengths. Based on Häni et al. (2021), the high $CH_4$ release rate of $140\,L_n\,min^{-1}$ was chosen in the present study to achieve a suitable signal-to-noise ratio. This was generally accomplished, as the precision was 26 % of the median concentration enhancement for the OP-12$h$. In the nighttime release, where the highest recovery rates were determined, the concentration enhancement for OP-12$h$ was even larger and precision was generally below 23 %. Thus, to the best of our knowledge, biases in the intercalibration of the OP instruments or in the amount of released gas cannot explain the low IDM recovery rates. Therefore, it is likely that the deviations in the modelled dispersion by the applied bLS model from real conditions are mainly responsible for the lower IDM recovery rates.

Despite promising experimental conditions and careful execution of the experiment, such as long measurement paths (the use of path-integrated concentration measurements is much less sensitive to biases in the wind direction than point measurements; Häni et al., 2024), relatively long fetches, high release rates, and the terrain being approximately horizontally homogeneous and flat, the recovery rates were lower than expected. In our case, the $CH_4$ was actively released in-

side the barn about 1.5 m above ground and might have left the barn at an even greater height above ground; thus, the initial vertical displacement of the $CH_4$ could have led to lower emission estimates, since the bLS model assumes a diffusive ground source. Also, the flow distortion caused by the barn and the nearby tree could have led to an updraft resulting in increased vertical mixing of the plume. To verify this hypothesis, a vertical profile of the $CH_4$ concentrations inside the plume would provide insight on changes in the recovery rates with height.

## 5  Conclusions

The median IDM recovery rates of the release experiment were 0.55–0.75 and thus smaller than 1, which cannot be explained conclusively. We hypothesise that the barn and the tree in the main wind axis led to the systematic underestimation of the derived emission rates due to the deviations in the wind field and to turbulent dispersion from the ideal assumptions in the bLS model. However, information regarding the shape of the plume was not available. It is important to note that the present study does not provide conclusive evidence that the IDM with the bLS model applied generally underestimates barn emissions. In our study at a fetch of 12$h$, we were still in the disturbed zone from the barn and the tree. Other studies using the applied bLS model have shown nearly 100 % recovery with comparable fetches for similar release experiments or good agreement with an independent reference method. Thus, there is no universally valid minimum distance at which one must place the concentration measurements downwind of a source to obtain accurate results. More experiments with controlled gas releases (including the tracer ratio method) inside a barn would be desirable for validation. Additional downwind vertical profile measurements of the concentration might help to detect deviations in shape of the dispersion plume.

*Data availability.*  Data from the MC, IC1, and IC2 can be accessed at https://doi.org/10.5281/zenodo.13218739 (Bühler, 2024).

*Supplement.*  The supplement related to this article is available online at: https://doi.org/10.5194/amt-17-1-2024-supplement.

*Author contributions.*  TK was responsible for funding acquisition. MB, CH, and TK were responsible for conceptualisation. MB and PB were responsible for conducting the $CH_4$ release. MB and CH were responsible for data evaluation. MB was responsible for the visualisation. MB was responsible for writing the original draft, with essential inputs from CH, AN, CA, and TK.

*Competing interests.* At least one of the (co-)authors is a member of the editorial board of *Atmospheric Measurement Techniques*. The peer-review process was guided by an independent editor, and the authors also have no other competing interests to declare.

ther geographical representation in this paper. While Copernicus Publications makes every effort to include appropriate place names, the final responsibility lies with the authors.

*Acknowledgements.* We thank the owner of the barn and the farmers of the land in the surrounding areas for their collaboration and assistance. We thank Simon Bowald for the planning of the $CH_4$ source, and we thank Simon Bowald and Martin Häberli-Wyss for their support during the measurements (both from the School of Agricultural, Forest and Food Sciences, Zollikofen).

*Financial support.* This research has been supported by the Bundesamt für Umwelt (grant no. 00.5082.PZ/BECDD68E6).

*Review statement.* This paper was edited by Huilin Chen and reviewed by two anonymous referees.

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

**Remarks from the language copy-editor**

CE1      Please note the slight changes to the wording of this new text.

**Remarks from the typesetter**

TS1      Please check: in Table 1, you asked to change "3.3" to "3.4", however this sentence is using "3.3". Please clarify.

TS2      Thank you for your explanation. We will start the post-review adjustments process as soon as the question regarding the value in the main text has been clarified.

TS3      Thank you for your explanation. We will start the post-review adjustments process as soon as the question regarding the value in the main text has been clarified.