# Peer review of "Applicability of the inverse dispersion method to measure emissions from animal housings"

_Atmospheric Measurement Techniques, 2023_

## Author Comment (AC1)

**Answers to reviewer comments RC1 from Anonymous Referee #2 on amt-2023-258**

**General Comments**

The simplicity of an idealized IDM technique for calculating gas emissions (relative to other approaches) makes the technique highly valuable, particularly if it can be accurately applied in situations that do not meet the "idealized" standard of horizontally homogeneous wind flow. This study looks at one such situation: emissions from animal barns. It examines the accuracy of an IDM calculation of barn emissions (Q) versus the distance from the barn where concentration (C) is measured. The idea is that while the C-Q relationship is affected by the wind complexity near the barn that results in IDM errors, as the distance of the C measurement increases the IDM error is reduced. At some threshold distance D from the barn, the idealized IDM calculation becomes suitably accurate. Exploration of the threshold D is useful given the wide range of emission sources accompanied by some type of wind complexity. The subject matter is suitable for AMT and will be a valuable contribution to efforts to understand the limitations of IDM (and how IDM can be used to provide accurate results).

We thank the Referee #2 for the positive feedback.

This study follows from a handful of others having similar objectives. Earlier work has suggested that D depends on the height of obstacles (h) around the source (e.g., barn height or fence height). In this manuscript the authors conduct a tracer release study and found that for the range of downwind locations examined (D/h < 21 or < 12) that Q was biased to underestimate the true gas release rate. I have some general comments about the presentation and interpretation of the results. These are:

1. Previous studies suggested the threshold distance D (for idealized-IDM accuracy) relates to the dominant height h of obstacles around the source. These obstacles could be barns, trees, or fences. The idea being that the wind flow disturbance extends over a downwind distance scaling with h. In this experiment, the largest obstacle seems to be the large tree beside the barn. If the authors want to compare their results with earlier studies, they should focus on D scaled with the tree height (not the barn height). The authors are aware of this, but it should be stated more clearly.

We agree with the reviewer and revised the manuscript accordingly. All instrument distances are now indicated as ratio of the tree height divided by the distance to the tree (i.e., the tree next to the barn, which was the highest obstacle at the site).

2. The basis for arguing for a threshold distance D (for idealized-IDM accuracy) is that the wind disturbance caused by an obstacle has limited spatial extent, and far downwind of the disturbed zone the resulting gas plume will become indistinguishable from the plume exposed to undisturbed flow, so at some distance dispersion can be accurately calculated based on the undisturbed wind. In this thinking the proper wind measurement location is upwind of the disturbance (or very far downwind). Measurement locations in the disturbed zone have no purpose in this paradigm. In this study the authors measure wind in the disturbed zone and treat these as another valid option for use in IDM. I would like to see the authors recognize these locations do not fit the hypothesis espoused in earlier

studies. I am not requesting they be eliminated, but they should not be taken as "equal" to the upwind measurements (unless the authors want to describe a different IDM paradigm).

*We followed the suggestion of the reviewer and only present the recovery rates obtained with the upwind turbulence measurements in the main manuscript. The other recovery rate results determined with data from the 3D ultrasonic anemometers placed downwind of the source were moved to the supporting information (SI-7).*

3. The authors published an earlier paper critical of the accuracy of the Boreal GF3 laser for measuring gas concentration. It would be good to see comments reconciling this earlier critique with the use of GF3's for this study. I am concerned about the inaccuracy of the GF3's for this study. I am not convinced by authors broad statement that laser calibration can be excluded as a cause of Q error.

*This is a valid point from the reviewer. We added Section 3.1, which addresses the precision of the OP estimated from the on-site intercomparisons. Further, we provide the concentration enhancements during the two release phases in Section 3.2. This allows a better assessment of whether the OP measurements were responsible for the low recovery rates (Section 4.3).*

4. The section on "Plume modeling and wind field rotation" has limited usefulness and I suggest it be eliminated. The wind field downwind of the barn is clearly complicated and the barn plume structure will be impacted by that complexity. This is not represented in the IDM calculations of the plume maps. Using different anemometer locations in the different maps calculations (and then assuming a horizontally uniform field) does not address the problem – all the calculations are wrong in detail. There is little to be learned with these plume maps and they are deceptive. This material is not critical to the main objective.

*We agree with the reviewer's comment and eliminated this section from the manuscript.*

5. The main results of this study (in terms of elucidating a threshold D for IDM) are broadly consistent with earlier work, and I would like the authors to more clearly state this (i.e., in the conclusions). Earlier studies suggest that D/h ranges from 5 to 30 over a range of circumstances. This study suggests D/h > 12. This broadly fits within earlier work. There is no reason to expect a universal D/h value ("Moving 10h (or any specific distance) downwind of an obstacle is unlikely to be a universal threshold for ignoring wind disturbances …"; Flesch et al. 2005, Deducing ground-to-air emissions from observed trace gas concentration, J Applied Meteorol). It would be good to more clearly place the study results into these earlier lines of thinking.

*We followed the reviewer's suggestion and introduced this issue in the Discussion (Section 4.3) and Conclusions.*

**Specific Comments**

6. Ln 29: "*The IDM is a micrometeorological method that combines concentration measurements up- and downwind of the spatially defined source with an atmospheric dispersion model*". IDM is more flexible than described here. The input concentration measurements do not have to be an upwind-downwind pair. Bai et al. (2023, Measurement of long-term CH4 emissions and emission factors from

beef feedlots in Australia, Atmosphere) is an example where vertically separated concentration measurements above the source were used to calculate emissions.

*This sentence was changed accordingly to be less specific, reflecting the different ways IDM can be done (lines 31-32).*

7. Ln 31: "*The IDM with a bLS model has been verified in multiple release experiments on open fields that reflect ideal conditions in terms of Monin-Obukhov-Similarity theory …*". It would be good to remind the audience of situations when MO similarity theory should be theoretically accurate, e.g., a horizontally homogeneous surface layer, where the source-to-sensor distance would be less than order 1 km.

*We added a sentence which specifies this point (lines 35-36).*

8. Ln 103: " … *factory calibration were applied* …". The authors previously published a very interesting paper on problems with the Boreal GF3 lasers used in this study: e.g., "*Application with paired devices needs an intercalibration of the devices. However, it remains unclear to what extent a side-by-side intercalibration can be transferred to the actual measurement setup, since relocation of the devices might cause systematic changes, as indicated by the different regression coefficients for different intercomparison campaigns.*" I would like the authors to comment on the capability of the GF3 to accurately measure CH4 concentrations in this study, in the context of their earlier criticisms.

*We added Section 3.1, which addresses the precision of the OP and the measured concentration enhancements. E.g. the precision of OP-12h was in the measurement campaign 6.3 ppm-m whereas the median concentration enhancement during the $CH_4$ release phase was 22.1 ppm-m.*

9. Section 3.3. "*Plume modeling and wind field rotation*". As mentioned earlier, I think this section has limited usefulness. There is little to be learned from the plume maps, and I believe they are deceptive. I would delete. I suggest a simpler example to illustrate the wind complexity (what about a simple wind vector plot of the wind measurement locations for a small number of periods?).

*The entire section was deleted.*

10. Ln 219: The authors discuss how the wind varies downwind of the barn, and reference earlier recommendations regarding how far an IDM concentration measurement should be made downwind of a barn. These earlier recommendations were made based on the "barn or other dominate obstacle height". It is perhaps unfair to apply these earlier recommendations based on barn height – earlier authors would argue the larger tree is the dominate obstacle height.

*We agree with the reviewer and use the tree height as dominant obstacle height throughout the manuscript.*

11. Ln 241: "*A bias in the results due to biases in the intercalibration of the OP or in the amount of released gas could be excluded. When the barn was excessively vented after the CH4 release, no*

*increase in the downwind CH4 concentration could be observed, indicating that no CH4 was kept back inside the barn*." This conclusion is too strong. A bias in the absolute concentration measurement (of all lasers) would certainly lead to biased estimates of the emission rate. Errors in gas release rate would similarly bias the results.

We only partly agree. As the OP are intercalibrated using the offset and the span, a bias (offset) in the absolute concentration should not lead to biased estimates of the emission rate. However, a bias in the slope correction could for sure lead to a bias in the recovery rates and this we can not fully exclude. With the findings from Section 3.1 we reformulated these sentences.

**Technical Corrections**

12. Ln 82: "… *and a 20 m long with* …" should this be a "20 m long *tubing* with …"?

Was corrected as suggested (line 86).

13. Ln 107: "*The concentrations between the five OP were inter-calibrated* …" How were the lasers inter-calibrated? Did the regression fit only a multiplier (slope) or a slope & offset?

This sentence was extended, and it states now that both, the slope and the offset were used (lines 110 – 111). We further added information that we missed out in the first submission (line 112-113).

14. Ln 124: For the anemometer placed upwind of the barn … what was the distance from the anemometer to the closest upwind obstacle (trees? building? Equipment?), both in absolute distance and distance scaled with the obstacle height?

There were no obstacle between the anemometer and the dam with trees in the north. The distance was 370 m. Assuming that the trees were about 25 m height, this would correspond to a fetch of >12h. But the height of the trees were only guessed based of pictures taken and not measured.

15. Ln 137: "… *the concentration upwind of the source is equally measured.*" What does "equally" mean here? Maybe it should be "also" measured?

'equally' was changed to 'also'.

16. Ln 222: "… *if the 15 m high tree is considered as the relevant flow disturbance* …" Because the tree is located southwest of the barn, is the "fetch" the distance from the tree to the downwind laser, or the distance from the barn? Clarify.

Now, we are using the tree height as dominant obstacle height throughout the manuscript.

---

## Author Comment (AC2)

**Answers to reviewer comments RC2 from Anonymous Referee #1 on amt-2023-258**

**General comments:**

This article on the applicability of the inverse dispersion method for measuring emissions from animal housing is exemplary in its writing and argumentation. The clarity and coherence of the content make it a valuable contribution to the field. However, the review identifies two notable drawbacks – (1) the challenge of frequently suboptimal atmospheric conditions and the limited range of conditions considered, (2) the use of the threshold distance/fetch which is usually based on the dominant obstacle height (rather than the source height). Concerning the latter one, I would argue that the tree (15 m) is the dominant wind obstacle rather than the barn itself. The discussion (chapter 4) would be further enriched by incorporating more literature, providing a more comprehensive context for readers. Despite these considerations, the article stands out as a well-crafted and insightful exploration of emission measurement methods.

We thank the Referee #1 for the feedback. We would like to comment the three points mentioned.

(1). For Swiss conditions, the atmospheric conditions were quite optimal. Often there are only low wind speeds ($u < 1$ m s$^{-1}$). The release phases cover stable, near neutral and unstable atmospheric conditions and thus the main range of conditions. Having more data would be preferable, however, the project budget did not allow to release methane for a longer time.

(2). We agree with the reviewer and have related the fetch to the tree height throughout the manuscript.

(3). Further literature was added in Section 4.3 to place our study into these earlier lines of thinking and providing more context for the readers.

**Specific comments:**

line 25: The introduction could benefit from more recent literature (e.g. instead of Stocker et al., 2013 and Gerber, 2013)

We updated the IPCC technical summary of AR5 with AR6 and added an additional source (lines 24-26). However, we think that Gerber et al, (2013) is still a relevant source and gives a good overview on the topic.

line 100: please provide further information on the device specific relationships as it is important for the accuracy of the concentration measurements

We added a Section 3.1 about the precision of the OP and the concentration enhancements during the release.

line 110: please define the input parameters for the model

We added all input parameters to Section 2.6.

lines 120/145/215/240: An important scale to determine the distance between source and the downwind measurement location is the height of the largest wind obstacle. When comparing to other studies, it is essential to include the fetch based on the tree height rather than the barn height.

We agree with the reviewer and changed the distances of the instrument to fetches related to the tree throughout the manuscript.

Table 3: I find the term "All UA" and "All OP" misleading. Isn't it a mean/median value of the considered options?

This table was restructured, and the mentioned row and column were removed. It was the median, if all the data were considered and not the mean/median of the values given in the table.

Line 205: I appreciate the approach of a sensitivity analysis. But then other parameters should be considered as well (and not only the rotation of the wind direction).

Following the recommendations of Referee #2, the entire section was removed from the manuscript.

Line 225: Despite the influence of the barn and tree, the recovery rates did not substantially differ.

Based on the recommendations of Referee #2, only recovery rates from the ultrasonic anemometer placed upwind of the barn are shown in the main manuscript and the recovery rates determined with the UA located downwind of the barn and tree were moved to the supporting information SI-7. Thus, this sentence was removed from the manuscript.

Line 245: It may be worth to conduct a sensitivity analysis for several parameters instead of using only one – the rotation of the wind direction

We removed this part as it is too involved to accurately simulate sensitivity to different parameters for the given setup.

Line 275: Delete the second "with" in the sentence "Other IDM studies have shown …."

The second "with" was removed.